# Potential Toxic Effects of Exposure to Titanium Silicon Oxide Nanoparticles in Male Rats

**DOI:** 10.3390/ijerph19042029

**Published:** 2022-02-11

**Authors:** Shimaa Ghareeb, Didair Ragheb, Aly El-Sheakh, Mohamed-Bassem Ali Ashour

**Affiliations:** 1Laboratory of Pest Physiology, Plant Protection Research Institute, Zagazig Branch, Agricultural Research Center, Ministry of Agriculture and Land Reclamation, Giza 12618, Egypt; shimaa.ghareeb@gmail.com (S.G.); dr.elsheikh.ali@gmail.com (A.E.-S.); 2Molecular and Environmental Toxicology Research Laboratory, Plant Protection Department, Faculty of Agriculture, Zagazig University, Zagazig 44519, Egypt; dawasef@agri.zu.edu.eg

**Keywords:** TiSiO_4_ NPs toxicity, male rats, serum biomarkers, oxidative stress, genotoxicity, immune toxicity, hepatotoxicity, renal toxicity, reproductive toxicity, histological alterations, Si and Ti accumulation in rat tissues

## Abstract

Recently, nano titanium silicon oxide (TiSiO_4_ NPs) has been used in different fields and industries. Very few toxicological data exist for TiSiO_4_ NPs. In the present study, the potential adverse effects of oral exposure to a single dose of TiSiO_4_ NPs ≤ 50 nm (250 mg/kg b.w.) in adult male rats were investigated through the assessment of biomarkers for serum biochemical parameters, liver DNA damage, and histopathological examination and determination of Si and Ti in the exposed rat tissues. The results revealed that there were no significant changes in serum total protein, albumin, and triglycerides content, while total cholesterol level was significantly increased 7 days after exposure. TiSiO_4_ NPs significantly increased superoxide dismutase (SOD), glutathione peroxidase (GPx), acetylcholine esterase (AChE), lactate dehydrogenase (LDH) activity, luteinizing hormone (LH), and follicle-stimulating hormone (FSH) levels in the exposed rat serum, whereas alanine aminotransferase (ALT), aspartate aminotransferase (AST) activity, urea level, immunoglobulins (IgG and IgM) concentrations, progesterone, and testosterone levels were significantly decreased. The liver comet assay indices were significantly increased after 7 days post-exposure. Moreover, histopathological changes and the accumulation of Si and Ti in liver, kidney, spleen, and lung tissues of treated rats were recorded.

## 1. Introduction

Recently, nanoscience has grown tremendously with fast-expanding nanotechnology and its broader applications [1]. Nanotechnology is defined as the science of manipulating materials on extremely small dimensions. It also enables design, synthesis, and control at the nano length range (1–100 nm) [2]. Nanoparticles (NPs) are used in a wide variety of products such as industry, electronics, medicine, and agriculture. They are used as additives for paper packaging, paints, ceramics, foods, drug delivery, biosensors, and cancer therapy [3]. NPs are currently found in over 1000 items or product lines on the market [4]. They have a high surface area to volume ratio in contrast to the larger form of material because the ratio of surface to total atoms or molecules increases exponentially with decreasing particle size, which leads to high surface reactivity, and that affects their physical properties. The widespread application of nanoproducts has raised concerns about the risk on the environment and human being health because of their size and high reactive surface. The smaller size of NPs also imparts a different biokinetic behavior and ability to spread in more areas of the body [5]. Many studies reviewed the access of nanoparticles to the cell nucleus [6,7], and have been shown to have genotoxic potential, which may be due to interference with the structure or function of DNA repair enzymes or the direct interaction with the DNA and DNA-related protein [8,9].

Titanium silicon oxide nanomaterial (TiSiO_4_ NPs) have been used to improve a variety of products. It has been reported that the inclusion of nano titanium silicon oxide (TiSiO_4_ NPs) in the cellulose acetate (CA) ultrafiltration membrane matrix significantly improved the hydrophilicity, permeability, fouling properties, and thermal stability of these membranes and decreased the membrane hydraulic resistance [10]. Additionally, TiSiO_4_ NPs are used nowadays to produce antibacterial cotton fabrics. It was observed that cotton fabrics can gain antibacterial activity against Staphylococcus aureus and limited activity against *Escherichia coli* [11]. The impact of TiSiO_4_ NPs on soil biota to estimate a risk limit for this nano material has been previously studied. No significant effect on the reproductive function of earthworms and collembolans invertebrates was recorded. Nevertheless, significant phytotoxic data were registered at the growth of dicotyledonous plant species [12]. Aqueous suspensions of TiSiO_4_ were responsible for the mutagenic potential for TA98 and TA100 strains of *Salmonella typhimurium* after 30 days of soil incubation. The instability of TiSiO_4_ NPs after 2 h of soil incubation may have been responsible for their availability to yield toxic effects on *Salmonella typhimurium* [13,14].

It has been reported that TiSiO_4_ NPs have less cytotoxic and genotoxic effects compared to TiO_2_ NPs [15]. According to the WHO report [16], median lethal dose (LD_50_) of titanium dioxide (TiO_2_) for rats is more than 12 g/kg after oral consumption. However, many studies reviewed the toxic effect of titanium and silica nanoparticles in mammals after dosing. In mice and rats, TiO_2_ NPs induced hepatic and renal toxicity, biochemical disturbance, and abnormal changes in the different tissues [17,18,19,20,21]. It has been shown that ultrafine TiO_2_ can cause cyto- and genotoxicity and induce apoptosis in human lymphoblastoid and brain cells as well as produce reactive oxygen species (ROS), leading to toxicity [22,23,24,25].

Silica nanoparticles induced alternations in most of the biochemical markers, specifically, alkaline biomarkers for phosphatase activity, serum aminotransferase enzymes activities, LDH, total protein, total and direct bilirubin level, superoxide dismutase, malondialdehyde, glutathione reduced, glutathione peroxidase, glutathione reductase, and catalase [26,27,28]; induced pathological alterations in rat organs [29,30]; and caused a minor increase in DNA damage in the blood and liver of exposed rats [31]. Mesoporous silica NPs were absorbed into the intestinal tract and persisted in the livers of mice [32]. The liver, kidneys, spleen, and lungs were found to be the target organs when colloidal silica NPs were orally applied to rats [33]. 

Therefore, the present study aims to evaluate potential toxic effects of oral exposure to single dose of titanium silicon oxide nanoparticles (TiSiO_4_ NPs, ≤50 nm) in adult male rats using total protein, albumin, cholesterol, and triglycerides contents; oxidative stress (SOD and GPx); immune toxicity (IgG and IgM); neurotoxicity (AChE); hepatotoxicity (ALT, AST, LDH and total bilirubin); renal toxicity (creatinine and urea); reproductive toxicity (LH, FSH, progesterone and testosterone); genotoxicity (liver DNA damage (comet assay)); and histopathological changes, as well as accumulation of Si and Ti in rat tissues after dosing.

## 2. Materials and Methods

### 2.1. Chemicals and Reagents

TiSiO_4_ nanoparticles (Product No: 641731) were purchased from Sigma-Aldrich, Burlington, MA, USA. As specified by the supplier, the TiSiO_4_ NPs were ≤50 nm white powder of titanium and silicon oxide, with a purity of 99.8%, based on trace metal analysis. All commercial kits and other reagents and chemicals used were of analytical grade.

### 2.2. TEM Analysis and XRD Pattern of TiSiO_4_ NPs

A sample of the commercial TiSiO_4_ NPs used in the present study was suspended in deionized water and sonicated for 20 min, and TiSiO_4_ NPs suspensions were dropped on carbon-coated copper TEM grids and then dried prior to measurement. The morphology and size of the nanoparticles were characterized using transmission electron microscopy (TEM) Model JEM-2100, JEOL, Akishima, Tokyo, Japan. The crystalline and phase structures of the NPs were examined by X-ray diffractometer.

### 2.3. Animals and Treatments

Adult male Wistar rats, *Rattus norvegicus*, with an average weight of 120–140 g were purchased from the animal house of National Research Center, Giza, Egypt. The rats were maintained in plastic cages in an air-conditioned animal house (temperature 24 ± 1 °C 12:12 h, light–dark cycle) with free access to water and commercial standard food (pellet ration). The animals received a complete health ration throughout the experiment period.

#### 2.3.1. Toxicity Biomarkers and Histopathological Evaluation

Twenty-four male albino rats were divided into two groups; each group consisted of 12 animals that were allowed to acclimatize for 2 weeks before treatments. Group I served as untreated control and was given corn oil (0.5 mL/100 g rat b.w) through gavage. In group II, a single dose of 250 mg TiSiO_4_/kg body weight in corn oil (0.5 mL/100 g b.w.) was orally administered through gavage to each animal. The animals fasted overnight before administration, and they were prohibited from food for a further 3–4 h post-administration. At 7 and 28 days after dosing, rats were anesthetized with diethyl-ether by inhalation and were sacrificed. Blood samples were collected and prepared for determination of serum biochemical markers and organs were obtained for comet assay (liver) and histopathological examination (liver, kidneys, spleen, and lung). 

##### Serum Assays

The exuding blood samples were collected in clean centrifuge tubes and left at room temperature until clotted. After complete retraction of the clot, the samples were centrifuged under cooling at 5000 rpm for 15 min, and the serum was separated. Serum samples were transferred into 1.5 mL eppendorf tubes and then stored at –20 °C until use.

Total protein, albumin, cholesterol, and triglycerides:

Serum total protein, albumin, cholesterol, and triglycerides content were determined using Vitro Scient reagent kits, Belbes, Egypt (https://vitroscient.com/diagnostics/ (accessed on 13 December 2021)). Protein and albumin determination were carried out according to colorimetric endpoint methods [34,35], respectively. The enzymatic colorimetric methods that depend on Vitro Scient reagent kits were used to determine total cholesterol [36,37] and triglycerides [38]. Reagent preparations and assays were carried out according to the kit manufacturer’s protocol. The absorbance of standard (A standard) and specimen (A specimen) against reagent blank were determined at 570, 630 nm for protein and albumin, respectively, and 550 nm for cholesterol and triglycerides using semi-automatic biochemical analyzer, Secomam Basic spectrophotometer. The concentration of total protein and albumin was expressed as g/dL and mg/dL for cholesterol and triglycerides as follows: concentration (g/dL or mg/dL) = (Absorbance of specimen/Absorbance of standard) × Standard value. 

Oxidative stress markers:

Superoxide dismutase EC 1.15.1.1 (SOD) and Glutathione peroxidase EC 1.11.1.9 (GPx) levels were measured in the serum samples collected 7 and 28 days after administration using enzyme-linked immunosorbent assay (ELISA) systems (Rat Superoxide Dismutase SOD and Rat Glutathione peroxidase 1, GPX1 ELISA Kits, Cusabio Biotech Co., Ltd., Houston, TX, USA (Rat Glutathione peroxidase 1, GPX1 ELISA Ki. Available online: http://www.cusabio.com/ accessed on 13 December 2021. Reagent preparations and assays were carried out as mentioned by the kit manufacturer’s protocol. Standards or samples (10 ul) were added to the suitable microtiter plate wells pre-coated with an antibody specific to SOD or specific to GPx-1. Biotin-conjugated polyclonal antibody preparations specific for SOD or GPx-1, and Avidin conjugated to peroxidase (HRP) were added to the wells and incubated. Then, a TMB (3,3′5,5′ tetramethyl-benzidine) substrate solution was added to all wells. The wells that contained SOD/GPx-1 biotin-conjugated antibody, and enzyme-conjugated Avidin showed a color change. The enzyme–substrate reaction was terminated by the addition of a sulfuric acid solution, and the optical density of each well was determined at once spectrophotometrically; thus, the color change was estimated spectrophotometrically at 450 nm using RT-2100 C microplate reader, Rayto Life and Analytical Sciences Co., Ltd.,Shenzhen, China. The concentration of SOD or GPx in the samples was then evaluated by comparing the O.D. of the samples to the standard curve. SOD and GPx levels were expressed as U/mL and ng/mL, respectively.

Immune toxicity markers:

Serum Immunoglobulins IgG and IgM levels were measured using enzyme-linked immunosorbent assay (ELISA) systems (Rat immunoglobulin G (IgG) and M (IgM) ELISA Kits, Cusabio Biotech Co., Ltd., Houston, TX, USA (Rat immunoglobulin G (IgG) kit. Available online: http://www.cusabio.com/ accessed 13 December 2021. This assay employs the competitive inhibition enzyme immunoassay technique. Reagent preparations and assays were carried out as mentioned by the kit manufacturer’s protocol. Sample and standard (50 uL) were added into the microtiter plate wells pre- coated with a rat IgG or IgM. Horseradish peroxidase (HRP)-conjugated antibody specific for rat IgG or IgM was added to each microplate well and incubated. After a wash to remove any unbounded reagent, a substrate solution was pipetted to the wells, and the color generated was opposite to the amount of rat IgG or IgM in samples. The color development was ended, and the optical density of each well was determined within 5 min at 450 nm using the microplate reader mentioned above. IgG and IgM levels were expressed as ng/mL.

Neurotoxicity marker:

Acetylcholine esterase EC 3.1.1.7. (AChE) level was measured in the serum samples collected 7 and 28 days after administration using the quantitative sandwich enzyme-linked immunosorbent assay (ELISA) technique, Rat Acetylcholinesterase (AChE) ELISA Kit, cusabio Co., Ltd., Houston, TX, USA. Reagent preparations and assays were carried out according to the kit manufacturer’s protocol. Antibody specific for AChE was pre-coated onto a microplate. Samples and standards (100 uL) were added into the wells and each AChE present was bound by the immobilized antibody. After removing any unbounded substances, a biotin-conjugated antibody specific for AChE was pipetted to the wells. After washing, avidin-conjugated horseradish peroxidase (HRP) was pipetted to the wells. Any unbound avidin-enzyme reagent is removed by washing, the substrate solution was added to the wells, and the color was generated in proportion to the amount of AChE bound in the initial step. The color development is ended, and the intensity of the color is determined. Stop solution was added to each well and the optical density of the well was determined within 5 min at 450 nm using the microplate reader mentioned above. The enzyme level was expressed as pg/mL. 

Hepatotoxicity markers:

Serum levels of aspartate aminotransferase (AST), alanine aminotransferase (ALT), lactate dehydrogenase (LDH), and albumin (ALB), which are indices for liver function, were determined. 

Quantitative determination of alanine aminotransferase EC2.6.1.2 (ALT) and aspartate aminotransferase EC2.6.1.1 (AST) were conducted using NADH kinetic UV method (Spinreact ALT and AST kits, Girona, Spain. Reagent preparations and assays were carried out as mentioned by the kit manufacturer’s protocol. A 100 μL sample was added to 1 cm light path cuvette containing 1 mL working reagent and then mixed well and incubated for 1 min. Initial absorbance (A) of the sample was read at 340 nm using a semi-automatic biochemical analyzer, Secomam Basic spectrophotometer, and then read again after 1, 2, and 3 min. Finally, the difference between absorbances and the average absorbance differences per minute (∆A/min) was calculated, and the enzyme activity was determined: activity (U/L) = ∆A/min × 1750.

Lactate dehydrogenase EC 1.1.1.27 (LDH) activity was determined using Kinetic ultraviolet method (spectrum-diagnostics, Obour city, Cairo, Egypt, Lactate dehydrogenase kit. Available online: www.spectrum-diagnostics.com accessed on 13 December 2021. Working reagents and assays were carried out as mentioned by the kit manufacturer’s protocol. A 10 μL sample was added to a 500 μL working solution, mixed, and the initial absorbance was read after 30 s at 340 nm using the spectrophotometer mentioned above then read again after 1, 2, and 3 min. Finally, the mean absorbance change per minute (∆A/min) was determined. The enzyme activity was calculated: LDH activity (U/L) = 8095 × ∆A 340 nm/min.

Quantitative determination of total bilirubin in rat serum was carried out according to the diazo method of Jendrassik and Grof [39] using Vitro Scient bilirubin reagent kit, Vitro Scient, Belbes, Egypt. Available online: www.vitroscient.com (accessed on 13 December 2021). Reagent preparations and assays were carried out as mentioned by the kit manufacturer’s protocol. A 200 μL R1 (Sulfanilic acid 30 mmol/L/HCl 0.20 N), one drop R2 (Sodium nitrite 30 mmol/L), 1000 μL R3 (Caffeine 0.26 mol/L/Sodium benzoate 0.52 mol/L), and 200 μL specimen were mixed, and then it was left to stand 10 min at room temperature. After that, R4 (Tartarate 0.93 mol/L/Sodium hydroxide1.90 N) was added, mixed, and then incubated for 5 min at room temperature. The obtained green color was stable for 30 min. The absorbance of specimen (A specimen) was determined against specimen blank (without R2) at 578 nm using the spectrophotometer mentioned above. The bilirubin concentration was estimated using the following formulae: Total Bilirubin (mg/dL) = Specimen absorbance × 10.8.

Renal toxicity markers:

Creatinine was determined using Vitro Scient Creatinine reagent kit, Belbes, Egypt. Available online: www.vitroscient.com (accessed on 13 December 2021). Working solutions and assay were carried out as mentioned by the kit manufacturer’s protocol. Trichloroacetic acid (1 mL) and 1 mL serum were mixed and centrifuged at 2500 rpm for 10 min, and then the supernatant (protein free filtrate (PFF)) was collected. For blank, 0.5 mL distilled H_2_O, 0.5 mL Trichloroacetic acid 1.2 M, and 1 mL working solution were mixed; for standard, 0.5 mL standard, 0.5 mL Trichloroacetic acid 1.2 M, and 1 mL working solution were mixed; while for the specimen (PFF), 1 mL PFF and 1 mL working solution were mixed. Finally, the absorbance of specimen and standard against blank (after 20 min) were determined using the spectrophotometer mentioned above. The concentration of creatinine was calculated using the following formula: creatinine (mg/dL) = (Absorbance of specimen/Absorbance of standard) × Standard.

Quantitative determination of urea in serum was carried out according to the enzymatic, colorimetric method (urease) modified Berthelot reaction [40] using Vitro Scient UREA/BUN reagent kit, Belbes, Egypt. Available online: www.vitroscient.com (accessed on 13 December 2021). Working solutions and assays were carried out as mentioned by the kit manufacturer’s protocol. The absorbance of the specimen (A specimen) and standard (A standard) against reagent blank was determined at 630 nm using the spectrophotometer mentioned above. The concentration of urea was calculated as follows: Urea (mg/dL) = (Absorbance of specimen/Absorbance of standard) × Standard value.

Reproductive toxicity markers:

Follicle-stimulating hormone (FSH) and luteinizing hormone (LH) levels were measured using rat FSH and rat LH enzyme-ELISA kits, Cusabio Biotech Co., Ltd., Houston, TX, USA. The assay employed the competitive enzyme immunoassay technique. The microtiter plate provided in the kit was pre-coated with goat-anti-rabbit antibody. Reagents and assay were carried out according to the kit manufacturer’s protocol. Samples and standards (50 uL) were pipetted to the microtiter plate wells with an antibody specific for (FSH or LH) and horseradish peroxidase HRP conjugate (FSH or LH). The competitive inhibition reaction was initiated between HRP labeled (FSH or LH) and unlabeled (FSH or LH) with the antibody. A substrate solution was pipetted to the wells, and therefore, the color was generated opposite to the quantity of (FSH or LH) within the sample. The color development was ended, and the intensity of the color was determined. The optical density of each well was determined within 10 min at 450 nm using RT-2100 C microplate reader, Rayto Life and Analytical Sciences Co., Ltd. FSH and LH concentrations were expressed as mlU/mL.

Quantification of testosterone and progesterone in serum was carried out using Cayman’s Testosterone ELISA Kits, Cayman chemical, Ann Arbor, MI, USA. Testosterone ELISA Kit. Available online: www.caymanchem.com (accessed on 13 December 2021). The assay depends on the competition between testosterone or progesterone–acetylcholinesterase (AChE) conjugate (testosterone or progesterone tracer) and testosterone or progesterone for a limited quantity of testosterone progesterone or progesterone antiserum. Reagents and assay were carried out as mentioned by the kit manufacturer’s protocol. This antiserum–testosterone or progesterone complex bound to mouse monoclonal anti-rabbit IgG that had been previously added to the well. The plate was washed to exclude any unbound reagents, and then Ellman’s Reagent (which includes the substrate to AChE) was pipetted into the well. The product of the reaction featured a clear yellow color and absorbed strongly at 412 nm. The intensity of the color was proportionate to the quantity of the Tracer bound to the well, which is inversely proportionate to the quantity of free testosterone or progesterone within the well during incubation. The testosterone and progesterone levels were expressed as ng/mL.

##### Genotoxicity (DNA Damage Marker)

Comet DNA assay:

Potential liver DNA damage was determined according to Singh et al. [41] using alkaline comet assay (single cell gel electrophoresis, SCGE). The liver was rapidly removed and quickly minced, suspended in chilled homogenization buffer (a 2 gm of a liver in 2 mL of cold Hanks Balanced Salt Solution (HBSS) containing 20 mM EDTA and 10% dimethylsulfoxide (DMSO), and then homogenized gently. Then, a 1.5 mL cell sample in micro-centrifuge tube was spun at 3500 rpm for 5 min at 5 °C. Liver homogenates (10 μL) were mixed with 90 μL of low melting point agarose, 0.7% in PBS, at 37 °C and added to a fully frosted microscope slide covered with 110 μL of normal melting point agarose (1% in PBS). A coverslip was directly placed on the top of the slide, and therefore, the agarose layer was left to solidify for 10 min (at 4 °C). The coverslip was then carefully removed, and another layer of low melting point agarose was added without cells slide. After that, the slides were covered by coverslip and kept at 4 °C for 5 min to let the agarose layer to solidify. After disposal of the coverslip, the slides were placed in lysis buffer (100 mmol/L Na2 EDTA; 2.5 Mol/L NaCl; 10 mmol/L; Tris, pH 10 with 10% DMSO and 1% Triton X-100 for at least 2 h at 4 °C. Then, the slides were placed within the electrophoresis chamber and incubated with electrophoresis alkaline buffer (1 mmol/L Na2 EDTA (pH > 13), 300 mmol/L NaOH) for 15 min at 4 °C to authorize for DNA uncoiling, and therefore, the appearance of alkali-labile DNA damage as strand breaks. Electrophoresis was performed for 30 min at 25 V and 300 mA. The slides were then washed 3 times, 5 min for each one, with a neutralization buffer (0.4 Mol/L Tris (pH 7.5)). Subsequently, slides were stained with 50 μL of ethidium bromide (2 mg/mL) and observed in an Optika Axioscope fluorescence microscope at 400× magnification. For each sample, 100 haphazardly chosen cells were photographed and scanned for image analysis. For each cell, the images were analyzed using the comet score analysis system. The length of DNA migration (tail length) was measured on PX from the center of the nucleus to the end of the tail. The DNA percentage in the tail was measured via the total intensity (fluorescence) in the cells, which was determined as 100%, and the percentage of total intensity paralleled to the intensity only in the tail was determined. The tail moment, shown in arbitrary units, was calculated as (tail length × % migrated DNA)/100.

##### Histopathological Examination of Liver, Kidneys, Spleen, and Lung Tissues in Male Rats Exposed Orally to TiSiO_4_

Liver, kidneys, spleen, and lung were dissected out and fixed in 10% formal saline (for 24 h). The samples were washed in tap water, dehydrated in ascending grade of ethanol, cleared in xylene, and embedded in paraffin (melting point 55–60 °C). Sections of 5 μm thickness were prepared and stained with haematoxylin and eosin. In this method, the paraffin sections were stained in Harris’s haematoxylin (for 5 min). Sections were washed in running water for bluing and then stained in 1% watery eosin (for 2 min), washed in water, dehydrated, cleared, and mounted in Canada balsam. The cytoplasm stained in shades of pink to red, and therefore, the nuclei gave blue color. Then, histopathological microscopic examination was carried out [42].

#### 2.3.2. Determination of Silicon and Titanium in Tissues of Male Rats Exposed Orally to TiSiO_4_

Eighteen male albino rats were divided into 3 groups. Each group consisted of 6 animals; 3 of them served as untreated control and were given corn oil (0.5 mL/100 g rat b.w.) through gavage, and the other three rats received one dose of 250 mg TiSiO_4_/kg b.w. in corn oil. The rats in the three groups were anesthetized with diethyl ether by inhalation, and then the rats were sacrificed at 1, 3, and 6 days after dosing, and the liver, spleen, kidney, and lung were collected. Silicon (Si) and titanium (Ti) contents in the tissues were determined according to [26]. The samples were dried and digested with nitric acid by microwave heating, and then the Si and Ti contents were determined using inductively coupled plasma optical emission spectrophotometer (ICP-OES). Si and Ti content was expressed as mg/kg tissue.

### 2.4. Data Analysis

The samples were assayed in triplicate, and the data were represented as the mean ± SD. All experimental values were compared with their corresponding control values. Differences between mean values were analyzed with one-way ANOVA by SPSS version 24 for Windows (IBM, Armonk, NY, USA). *p* < 0.05 were statistically significant.

## 3. Results

### 3.1. TEM Analysis and XRD Pattern of TiSiO_4_ NPs

A transmission electron microscopy (TEM) analysis image showed that ≤50 nm nanoparticles were the predominant entity present in the suspension with a relatively narrow size distribution and no evidence of aggregation of the TiSiO_4_ NPs evaluated (Figure 1). The XRD pattern of TiSiO_4_ NPs was graphically clarified in Figure 2. There were no peaks of impurities. The results confirmed that the NPs used in this study were TiSiO_4_ NPs, as the positions of all the diffraction peaks of the samples were consistent with the crystalline pattern of titanium silicon dioxide. Purity represents an important property of the test material.

### 3.2. Serum Biochemical Markers 

#### 3.2.1. Total Protein, Albumin, Cholesterol, and Triglycerides

Total protein, albumin, cholesterol, and triglycerides content in male rat serum at 7 and 28 days after oral exposure to 250 mgTiSiO_4_ NPs (≤50 nm)/kg b.w. are presented in Table 1. 

The obtained results revealed that there were no significant changes in total protein, albumin, and triglycerides content in male rat serum at 7 and 28 days after treatment compared with the control. Total cholesterol level was increased by 57.67% 7 days after exposure, while no significant changes were observed after 28 days.

#### 3.2.2. Oxidative Stress

SOD and GPx activity in male rat serum at 7 and 28 days after exposure to oral single dose (250 mg/kg b.w.) of TiSiO_4_ NPs, which are indices for oxidative stress, are presented in Table 2. The obtained data indicated that SOD and GPx activity was significantly increased as compared with the untreated controls (*p* < 0.05). The increase in SOD level was higher than GPx. SOD level was elevated by 225 and 108.2% at 7 and 28 days after exposure, respectively, whereas the corresponding increases for GPx activity were 165.28% and 91.44%, respectively. 

#### 3.2.3. Immune Toxicity 

Immunoglobulins G (IgG) and M (IgM) concentrations in male rat serum at 7 and 28 days after exposure to oral single dose (250 mg/kg b.w.) of TiSiO_4_ NPs are shown in Table 2. IgG and IgM levels were decreased significantly by 64.67 and 59.63% at 7 days after exposure, respectively, whereas the corresponding decreases at 28 days were 79.31 and 79.90% as compared with the untreated control (*p* < 0.05).

#### 3.2.4. Neurotoxicity 

Data in Table 2 show AChE activity in serum of male rats exposed orally to the single TiSiO_4_ NPs. AChE level was increased significantly, which reached 27.67 pg/mL and 37 pg/mL at 7 and 28 days post-administration, respectively. The corresponding control values were 11.88 pg/mL and 18.33 pg/mL (*p* < 0.05).

#### 3.2.5. Hepatotoxicity 

The results from the serum biochemical examination of the liver function biomarkers ALT, AST, LDH, and total bilirubin in male rats exposed orally to TiSiO_4_ NPs are presented in Table 3. The obtained data showed that there was no significant change in ALT activity in male rats at 7 days after exposure comparing with control (*p* < 0.05). At 28 days post-treatment, the activity was significantly decreased as it reached 108.70 U/L compared to 143.80 U/L in the control. The AST activity was significantly decreased by 49.91 and 44.89% at 7 and 28 days after dosing, respectively, compared with the control (*p* < 0.05). The LDH activity was significantly increased by 48.64% on day 7 after exposure and significantly decreased by 24.42% after 28 days post-administration compared with the control (*p* < 0.05). A significant increase in total bilirubin level (76.66%) was observed in TiSiO_4_ NPs exposed rats on day 7 after treatment, whereas there were no significant changes after 28 days post-exposure compared with the control (*p* < 0.05).

#### 3.2.6. Renal Toxicity

Creatinine and urea levels in rat serum, which are indices for kidney function, are presented in Table 3. The results showed that there were no significant differences between creatinine levels in the TiSiO_4_ NPs exposed male rats at 7 and 28 days post-administration and the control. Moreover, Urea level was decreased significantly by 17.28% on day 7 post-exposure, while there was no significant change after 28 days in treated rats compared with control (*p* < 0.05). 

#### 3.2.7. Reproductive Toxicity 

The levels of the biomarkers sex hormones FSH, LH, testosterone, and progesterone in the serum of male rats exposed to a single oral dose of TiSiO_4_ NPs are presented in Table 4. The obtained data revealed that FSH and LH concentrations were significantly elevated at 7 and 28 days after exposure. On the other hand, the levels of testosterone and progesterone were significantly decreased after 7 and 28 days as compared with the control (*p* < 0.05). The obtained changes in the evaluated sex hormones level may reflect a potential reproductive toxicity in the exposed male rats.

### 3.3. Genotoxicity 

The potential DNA damage in male rat liver exposed to single oral dose of TiSiO_4_ NPs (250 mg/kg b.w.) was determined using alkaline comet assay (single cell gel electrophoresis, SCGE). The assay is used as a biomarker for genetic toxicity. As shown in Table 5 and Figure 3, TiSiO_4_ NPs exposure caused a significant increase in comet percentage (98.79%), % DNA in the tail (59.97%), tail moment (192.85%), and olive tail moment (98.52%) in the treated male rats compared with the control on day 7 post-administration. There were no significant differences at 28 days after treatment.

### 3.4. Histopathological Examination 

Histopathological analysis of liver, kidney, spleen, and lung sections of male rats at 7 and 28 days post-treatment with single oral dose of TiSiO_4_ NPs (250 mg/kg b.w.) are shown in Figure 4, Figure 5, Figure 6 and Figure 7.

#### 3.4.1. Liver 

In the control group, liver sections showed normal hepatic cells with complete cytoplasm as well as intact nucleus, nucleolus, portal lobules, and central vein (Figure 4A,B). Microscopic examination of liver sections of the treated male rats at 7 and 28 days after treatment showed congested portal areas associated with necrosis of the surrounding hepatocytes (Figure 4C,D).

#### 3.4.2. Kidney 

Histological evaluation of section of kidney of control groups showed normal glomerulus and Bowman’s capsule intact with simple cuboidal epithelial lining. The proximal tubule and distal convoluted tubule were also normal (Figure 5A). In treated male rat kidney sections, at 7 days after exposure, most of the renal corpuscles appeared normal. Atrophy or partially degenerated of some glomeuli that associated with wide urinary space were found. The renal tubules appeared as control (Figure 5B). On day 28 after treatment renal corpuscles and renal tubules appeared normal (Figure 5C).

#### 3.4.3. Spleen 

Examination of the spleen sections from control male rats indicates the darkest region of the white pulp due to the presence of predominately small lymphocytes, the red pulp, and white pulp-containing lymphocytes (Figure 6A). A histology of spleen from treated male rats at 7 days post-administration exhibited a degenerated area with highly reduced white pulp (Figure 6B). Moreover, degenerated area in the white pulp disturbed architecture was observed after 28 days (Figure 6C).

#### 3.4.4. Lung 

Microscopic examination of sections of the lung of male rats from the control group showed normal alveoli that lined with type I pneumocyte squamous cells and type II pneumocyte cuboidal cells. Thin septa, alveolar sacs, and bronchiole were noticed (Figure 7A). In lung of treated male rats, some patent alveoli, moderate cellular infiltration in the septa and the adventitia of distended bronchiole, and moderately thickened septa were shown after 7 days post-administration (Figure 7B,C). However, alveoli that appeared with thin septa and alveolar sacs, normal bronchiole, thickened blood vessel wall, collapsed alveoli wall, heavy cellular infiltration in intra alveolar spaces, and markedly thickened septa were observed in lung sections of male rats at 28 days post-exposure (Figure 7D,E).

### 3.5. Silicon and Titanium Content in Tissues of Male Rats Exposed Orally to TiSiO_4_ NPs

Si and Ti content in the liver, kidney spleen, and lung of control and treated male rats with 250 mg TiSiO_4_ NPs/kg b.w after 1, 3, and 6 days post-oral administration is shown in Table 6 and Table 7 and Figure 8. Data in Table 6 and Figure 8 revealed that Si levels were detected in male rat kidney, lung, liver, and spleen tissues after 1, 3, and 6 days post-oral administration. In kidney and lung, significant higher values were obtained after 1, 3 and 6 days post-treatment as compared with the controls. In liver tissues, the significant increases were recorded at 1 and 3 days post-administration. Meanwhile, in spleen tissues, a significant increase in Si concentration was found only after 3 days post-treatment (87.14 mg Si/kg tissue) compared with the untreated control (60.60 mg Si/kg tissue).

Regarding titanium content, the results presented in Table 7 show that Ti was detected in liver and kidney 3 days after exposure. The titanium level was higher in kidney than in liver where the corresponding concentrations were 2.42 and 0.77 mg Ti/kg tissue. However, Ti concentrations in the rest treated and all control samples were below the level of detection (<0.05 mg Ti/kg tissue).

## 4. Discussion

Nowadays, the widespread application of nanomaterials has raised concerns regarding their risk on environmental and human health. To the best of our knowledge, few toxicological data exist for TiSiO_4_ NPs and no LD_50_, NOAEL or LOAEL values are published. The present study was carried out to evaluate the potential toxic effects of oral exposure to a single dose (250 mg/kg b.w.) of ≤50 nm TiSiO_4_ NPs in adult male rats after 7 and 28 days post-exposure. The animals’ behaviors and symptoms were carefully observed daily for 28 days post-administration. The treated male rats showed no death, abnormal behaviors, or symptoms as compared with untreated controls. Non-significant changes in total protein, albumin, and triglycerides level were recorded in the serum of treated rats, while total cholesterol level was significantly increased.

Our study indicates a significant increase in the antioxidant enzymes SOD and GPx; a significant decrease in IgG and IgM concentration; and a significant increase in AChE level. Elevated antioxidant enzymes indicate increase in oxidative stress. It has been reported that the decrease in TiO_2_ NPs-treated mice immunoglobulin occurs due to damage in the spleen and reduction in immune capacity [43]. In our study, the histopathology examination of spleen from TiSiO_4_ NPs-treated male rats showed a degenerated area with highly reduced white pulp. The suppression of humoral immunity by TiSiO_4_ NPs may be attributed to the degeneration and depletion in the white pulp of the spleen. The acetylcholine receptors (AChRs) modulate interactions among the nervous system and the immune system. So, close interactions exist between immunity and neurotransmission [44]. Since acetylcholine (Ach) is an anti-inflammatory molecule, elevated AChE concentration indirectly reflects a decreased concentration of Ach and an increase in the local and systemic inflammation [45].

In the current study, the activity of serum marker enzymes of the liver ALT and AST were significantly decreased, whereas LDH activity was significantly elevated in the treated rats. The alternation in serum levels of ALT, AST, and LDH activity may be indicating liver damage and alteration in its functions [46,47,48,49,50].

Moreover, serum total bilirubin level was increased in TiSiO_4_ NPs-treated male rats. Bilirubin is a major breakdown product of hemoglobin, increases when liver injured or damaged. The rise of total bilirubin outcomes from decreased uptake and conjugation of bilirubin by the liver due to liver cell dysfunction [51]. In this study, the histopathological findings in the liver of the TiSiO_4_ NPs- treated rats showed congested portal areas associated with necrosis of the surrounding hepatocytes. These results are correlated with those of oxidative stress and liver function markers. This finding indicates hepatic damage that alters hepatocytes membrane permeability, causing leakage of enzymes from the cells. When the tissues are subjected to injure, LDH leaked into the serum of blood from organs or cells. 

In our study, TiSiO_4_ NPs caused a significant increase in liver comet assay indices in the exposed male rats after 7 days post-administration. Meena et al. [15] reported that TiSiO_4_ has less severe cytotoxic and genotoxic effects on human embryonic kidney cells (HEK-293) compared to TiO_2_ nanoparticles. Several investigators have reported that TiO_2_ and SiO_2_ nanoparticles caused inflammation, oxidative DNA damage, and genotoxicity in treated mice and induced DNA damage by significant increases in % tail DNA and tail moment [52,53,54]. The mechanism for nanoparticles caused genotoxicity could be by intracellular oxidative stress and DNA damage that leads to cellular toxicity [55,56,57,58,59]. 

Furthermore, our data revealed no significant changes in serum creatinine level and a significant reduction in urea level in the kidney of TiSiO_4_ NPs-exposed rats. Histopathology of kidneys of the treated rats showed atrophy or partially degeneration of some glomeuli that associated with wide urinary space. The renal corpuscles and renal tubules appeared more or less normal. The decrease in urea levels indicated damage to nephrons. 

In the present investigation, progesterone and testosterone levels were significantly decreased in TiSiO_4_ NPs-treated rats, while LH and FSH levels were significantly increased. The changes in sex hormones level may reflect a potential reproductive toxicity in the exposed male rats.

In this study, total Si and Ti concentrations were determined in kidney, lung, liver, and spleen tissues of TiSiO_4_ NPs-treated and untreated male. Si levels were found to be significantly higher in kidney and lung after 1, 3, and 6 days post-administration as compared with untreated controls, whereas elevated Si levels were detected at 1 and 3 days in liver and 3 days in spleen. Increased Si levels in kidneys are often associated with the excretion pathway of the silica nanoparticles in urine [60]. Low Ti concentrations were detected only in liver and kidney tissues at 3 days after exposure. The detected level in kidney was higher than in liver. Titanium levels in treated and untreated controls were found to be below the detection limit in spleen, and lung tissues indicated a very low absorption. In oral administration, nanoparticles face stomach acid and cross the epithelium of the gastrointestinal tract to reach the blood circulation. Moreover, nanoparticles are subjected to metabolic processes that may affect their bioavailability [61].

Herein, it is worth clarifying that the resulting potential adverse effects on adult male rats are limited to oral exposure to a single high dose of TiSiO_4_ NPs (250 mg/kg b.w.). Further low dose–response and long-term exposure studies are needed to understand the changes in physicochemical properties of TiSiO_4_ nanoparticles and its molecular mechanisms by which it induces toxicity. Furthermore, detailed investigations on tissue distribution, excretion kinetics, and bioaccumulation/bioavailability of orally administered TiSiO_4_ NPs in rats are required.

## 5. Conclusions

Oral exposure of male rats to single dose (250 mg/kg b.w.) of ≤50 nm TiSiO_4_ NPs induces significant changes in biochemical markers, which are indices for oxidative stress, genotoxicity, neuro toxicity, hepatic toxicity, renal toxicity, reproductive toxicity, and immune toxicity. These changes were accompanied by histopathological changes in the tissues of the liver, kidney, spleen, and lung. However, the study was limited to the response to a single high oral dose of TiSiO_4_ NPs at two-time post-administration. Further low dose–response and long-term exposure studies are needed to evaluate the role of TiSiO_4_ NPs in potential toxicological effects and health hazards associated with oral exposure. Moreover, a metabolomic analysis approach could be used to study systemic effects of metabolites. Furthermore, investigations are required to determine the mechanism of harmful effects of TiSiO_4_ NPs in various cell lines and other animal species.

## Figures and Tables

**Figure 1 ijerph-19-02029-f001:**
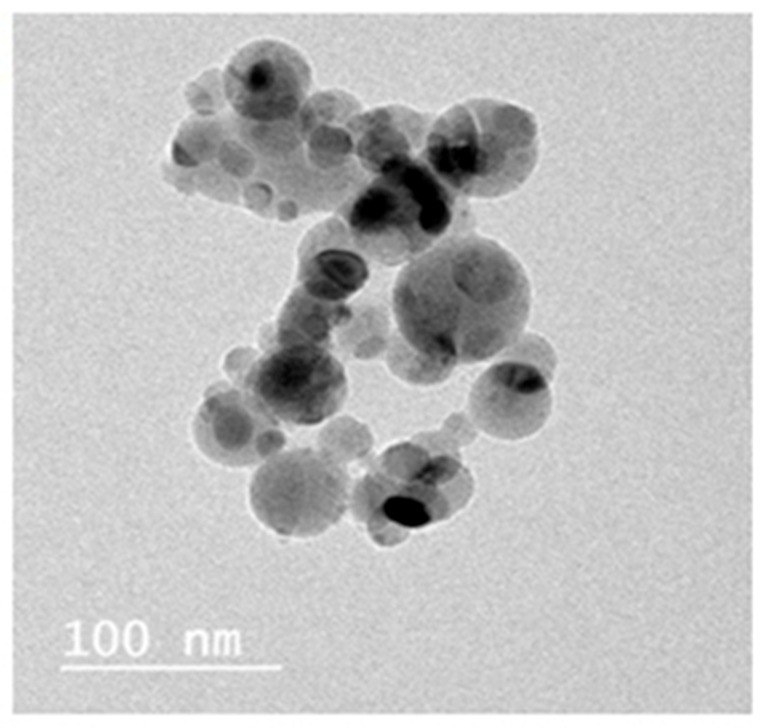
Transmission electron microscope images of TiSiO_4_ NPs.

**Figure 2 ijerph-19-02029-f002:**
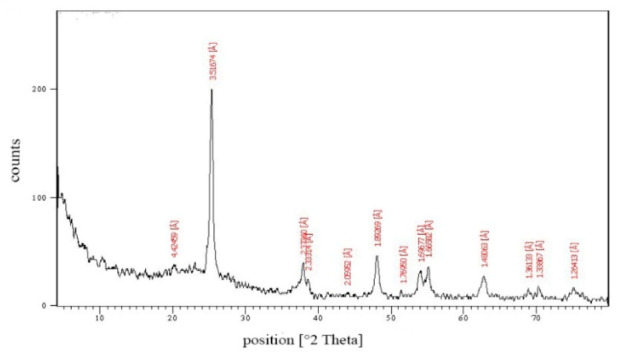
X-ray diffraction pattern of TiSiO_4_ NPs.

**Figure 3 ijerph-19-02029-f003:**
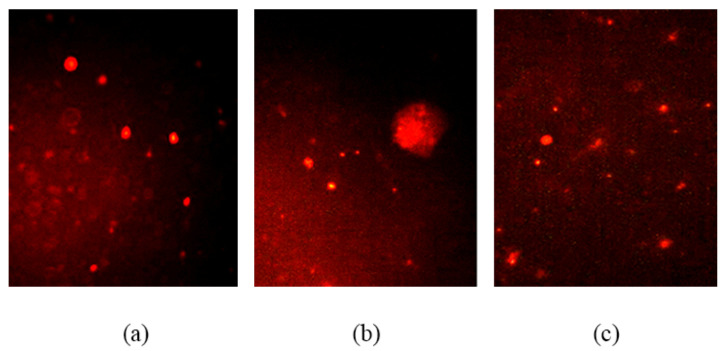
Fluorescence microscopy image of liver cell comets of male rats exposed to 250 mg TiSiO_4_ NPs/kg b.w.; (**a**) control, (**b**) 7 days post-treatment, and (**c**) 28 days post-treatment.

**Figure 4 ijerph-19-02029-f004:**
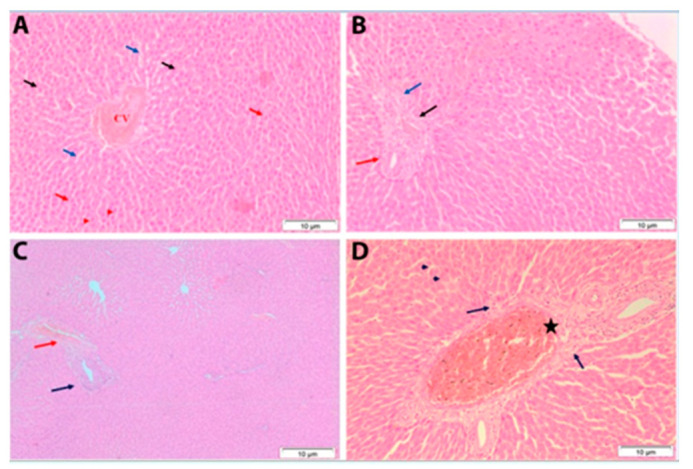
Histological analysis of the liver tissue in male rat exposed orally to 250 mg TiSiO_4_ NPs/kg b.w. (**A**,**B**): control liver showing normal structure, (**A**): normal hepatic lobule. The central vein (CV) lies at the center of the lobule and intact with the hepatocytes (black arrows) with strongly eosinophilic granulated cytoplasm (red arrows), and distinct nuclei (arrow heads). In between the strands of hepatocytes, the hepatic sinusoids are shown (blue arrows); (**B**): normal structure of portal area surrounded with connective tissue and each portal area contains a branch of hepatic artery (black arrow), a branch of portal vein (red arrow), and a bile ductile (blue arrow); (**C**): male rat liver after 7 days post-administration showing congested portal area (red arrow) associated with necrosis (blue arrow) of the surrounding hepatocytes; (**D**): male rat liver after 28 days post-treatment showing congested portal area (asterisk) associated with necrosis of the surrounding hepatocytes (arrows) and the remaining hepatocytes displayed the normal structure (arrow heads). (H & E stain, Scale bar: 10 μm).

**Figure 5 ijerph-19-02029-f005:**
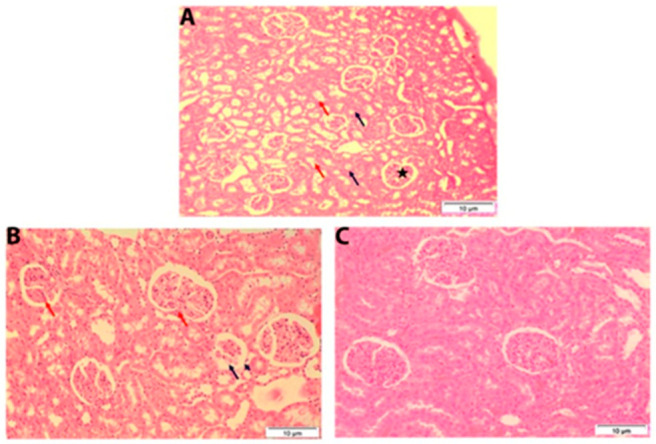
Histological analysis of the kidney tissue in male rat exposed orally to single dose of 250 mg TiSiO_4_ NPs/kg b.w. (**A**): Male rat kidney showing a normal glomerulus and Bowman’s capsule (asterisk) intact with simple cuboidal epithelial lining. The proximal tubule (blue arrow) and distal convoluted tubule (red arrow) are also normal. (**B**): Male rat kidney after 7 days post-administration showing that most of the renal corpuscles appeared normal. Notice atrophy (red arrow) or partially degenerated (blue arrow) of some glomeuli that associated with wide urinary space (arrowhead). The renal tubules appeared nearly as control. (**C**): Male rat kidney after 28 days post-exposure showing the renal corpuscles and renal tubules appeared more or less normal. (H & E stain, Scale bar 10 μm).

**Figure 6 ijerph-19-02029-f006:**
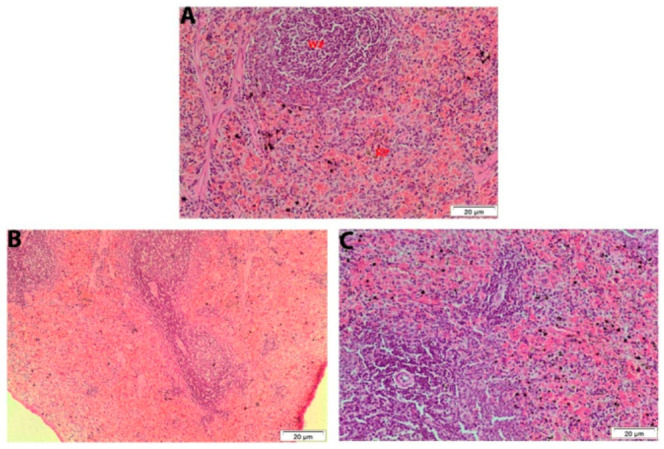
Histological analysis of the spleen tissue in male rat exposed orally to single dose of 250 mg TiSiO_4_ NPs/kg b.w. (**A**): Control male rat spleen indicates the darkest region of the white pulp (WP) due to the presence of predominately small lymphocytes. The red pulp (RP) and white pulp are containing lymphocytes. (**B**): Male rat spleen after 7 days post-administration showing degenerated area with highly reduced white pulp. (**C**): Male rat spleen after 28 days post-exposure showing degenerated area disturbed architecture. (H&E stain, Scale bar: 20 μm).

**Figure 7 ijerph-19-02029-f007:**
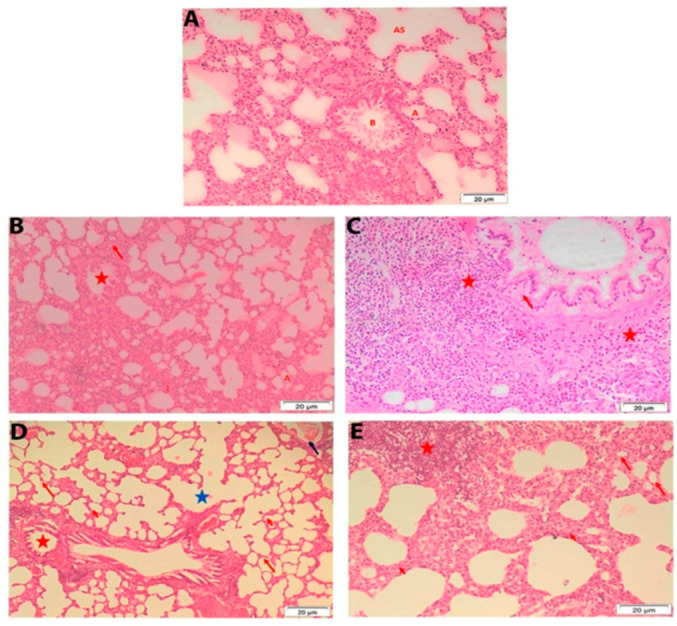
Histological analysis of the lung tissue in male rat exposed orally to single dose of 250 mg TiSiO_4_ NPs/kg b.w. (**A**): male rat lung showing normal alveoli (red A) that lined with type I pneumocyte squamous cells (P1), type II pneumocyte cuboidal cells (P2), thin septa (red I), alveolar sacs (red AS) and bronchiole (red B). (**B**,**C**) male rat lung after 7 days post administration showing (**B**): some patent alveoli (red A), moderate cellular infiltration (red arrow) in the septa and the adventitia of distended bronchiole, and moderate thickened septa; (**C**): some patent alveoli, moderate cellular infiltration (red asterisk) in the septa and the adventitia of distended bronchiole (red arrow), and moderate thickened septa (red arrowhead). (**D**,**E**) male rat lung after 28 days post administration showing (**D**): alveoli (red arrow heads) with thin septa (red arrows) and alveolar sacs (blue asterisk), normal bronchiole (red asterisk) and thickened blood vessels wall (blue arrow). (**E**): collapsed alveoli wall (red arrow), heavy cellular infiltration in intra alveolar spaces (red asterisk) and markedly thickened septa (red arrow heads).

**Figure 8 ijerph-19-02029-f008:**
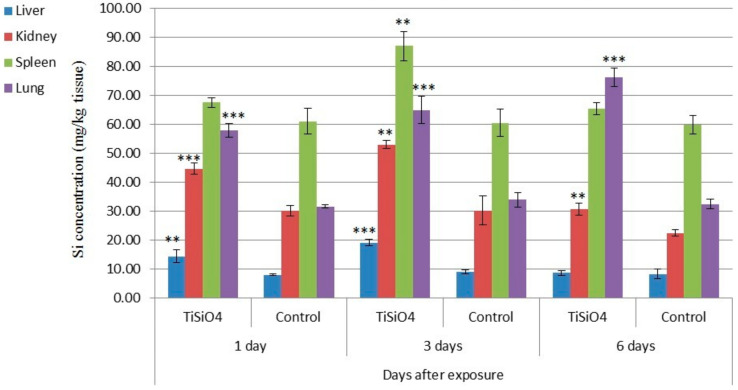
Silica content (mg/kg tissue) in male rat liver, kidney, and spleen tissues 1, 3, and 6 days post-oral administration of 250 mg ≤ 50 nm TiSiO_4_ NPs/kg b.w. Values are the mean ± SD **: moderate significance, ***: high significance (*p* < 0.05).

**Table 1 ijerph-19-02029-t001:** Serum total protein, albumin, total cholesterol, and triglycerides content in male rats exposed orally to 250 mg ≤ 50 nm TiSiO_4_ NPs/kg b. w.

Treatment	Days Post-Administration	Total Protein g/dL	Total Albumin g/dL	Total Cholesterol mg/dL	Triglycerides mg/dL
Control	7 Days	6.17 ± 0.68	4.07 ± 0.59	94.0 ± 3.00	66.0 ± 17.00
TiSiO_4_ NPs		6.53 ± 0.15	3.63 ± 0.06	151.67 ± 34.03	87.67 ± 5.51
*p*		0.4141 ^ns^	0.2714 ^ns^	0.0431 *	0.1036 ^ns^
Control	28 Days	6.0 ± 0.40	3.27 ± 0.12	82.0 ± 15.72	44.00 ± 4.36
TiSiO_4_ NPs		5.63 ± 0.29	3.33 ± 0.15	85.0 ± 19.00	55.33 ± 7.37
*p*		0.2674 ^ns^	0.5790 ^ns^	0.8434 ^ns^	0.0836 ^ns^

Note: Values are the mean ± SD, *: low significance, ^ns^: not significant. (*p* < 0.05).

**Table 2 ijerph-19-02029-t002:** Oxidative stress, immunotoxicity, and neurotoxicity in male rats exposed orally to 250 mg ≤ 50 nm TiSiO_4_ NPs/kg b. w.

Treatment	Days Post-Administration	Oxidative Stress Markers		Immunotoxicity Markers		Neurotoxicity Marker
		SOD U/mL	GPx ng/mL	IgG ng/mL	IgM ng/mL	AChE pg/mL
Control	7 Days	8 ± 1.00	8.67 ± 2.08	2508.00 ± 108.00	553.33 ± 54.52	11.88 ± 2.26
TiSiO_4_ NPs		26 ± 2.00	23.00 ± 3.61	886.00 ± 38.00	223.33 ± 23.35	27.67 ± 1.53
*p*		0.0002 ***	0.0040 **	0.0000 ***	0.0006 ***	0.0006 ***
Control	28 Days	16.33 ± 4.73	15.67 ± 1.53	1637.00 ± 133.64	358.33 ± 28.99	18.33 ± 1.53
TiSiO_4_ NPs		34 ± 2.00	30.33 ± 1.53	338.67 ± 48.50	72.00 ± 8.00	37.00 ± 3.00
*p*		0.0040 **	0.0003 ***	0.0001 ***	0.0001 ***	0.0007 ***

Note: Values are the mean ± SD, **: moderate significance, ***: high significance. (*p* < 0.05).

**Table 3 ijerph-19-02029-t003:** Hepatic and renal toxicity in male rats exposed orally to 250 mg ≤ 50 nm TiSiO_4_ NPs/kg b.w.

Treatment	Days Post-Administration	Hepatic Function Markers				Renal Function Markers	
		ALT U/L	AST U/L	Bilirubin mg/dL	LDH U/L	Creatinine mg/dL	Urea mg/dL
Control	7 Days	105.19 ± 13.26	250.18 ± 14.94	0.30 ± 0.10	1142.33 ± 24.50	0.53 ± 0.06	81 ± 0.00
TiSiO_4_ NPs		64.60 ± 22.54	125.31 ± 10.68	0.53 ± 0.06	1698.00 ± 177.00	0.63 ± 0.06	67 ± 6.0
*p*		0.0547 ^ns^	0.0003 ***	0.0249 *	0.0057 **	0.1012 ^ns^	0.0156 *
Control	28 Days	143.80 ± 10.71	174.48 ± 11.26	0.33 ± 0.06	1145.00 ± 126.50	0.57 ± 0.06	71 ± 8.50
TiSiO_4_ NPs		108.70 ± 6.60	96.15 ± 7.83	0.43 ± 0.06	865.33 ± 52.50	0.53 ± 0.06	55 ± 12.12
*p*		0.0084 **	0.0003 ***	0.1012 ^ns^	0.0241 *	0.5185 ^ns^	0.1287 ^ns^

Note: Values are the mean ± SD, *: low significance, **: moderate significance, ***: high significance, ^ns^: not significant (*p* < 0.05).

**Table 4 ijerph-19-02029-t004:** Sex hormones (FSH, LH, testosterone, and progesterone) levels in serum male rats exposed orally to 250 mg ≤ 50 nm TiSiO_4_ NPs/kg b.w.

Treatment	Days Post-Administration	Sex Hormones			
		LH mlU/mL	FSH mlU/mL	Testosterone ng/mL	Progesterone ng/mL
Control	7 Days	0.65 ± 0.18	0.45 ± 0.09	0.44 ± 0.02	1.15 ± 0.25
TiSiO_4_ NPs		5.16 ± 0.95	2.71 ± 0.39	0.07 ± 0.01	0.59 ± 0.18
*p*		0.0012 **	0.0006 ***	0.0000 ***	0.0326 *
Control	28 Days	1.22 ± 0.22	0.95 ± 0.15	0.37 ± 0.08	1.26 ± 0.27
TiSiO_4_ NPs		13.14 ± 3.05	4.53 ± 0.76	0.08 ± 0.01	0.42 ± 0.03
*p*		0.0025 **	0.0013 **	0.0024 **	0.0054 **

Note: Values are the mean ± SD, *: low significance, **: moderate significance, ***: high significance (*p* < 0.05).

**Table 5 ijerph-19-02029-t005:** Comet assay indices in liver male rats exposed orally to 250 mg ≤ 50 nm TiSiO_4_ NPs/kg b.w.

Treatments	Days Post-Administration	% Comet	Tail Length (px)	%DNA in Tail	Tail Moment	Olive Moment
Control	7 Days	8.30 ± 0.88	5.76 ± 1.12	7.37 ± 1.85	0.56 ± 0.13	0.68 ± 0.20
TiSiO_4_ NPs		16.5 ± 0.80	8.54 ± 1.86	11.79 ± 1.31	1.64 ± 0.46	1.35 ± 0.25
*p*		0.0003 ***	0.0900 ^ns^	0.0278 *	0.0174 *	0.0141 *
Control	28 Days	8.45 ± 0.55	6.69 ± 0.63	8.38 ± 1.49	0.88 ± 0.13	0.82 ± 0.11
TiSiO_4_ NPs		8.7 ± 1.05	6.74 ± 0.84	8.47 ± 1.03	0.91 ± 0.24	0.94 ± 0.16
*p*		0.7340 ^ns^	0.9380 ^ns^	0.9404 ^ns^	0.8400 ^ns^	0.3576 ^ns^

Note: Values are the mean ± SD, *: low significance, ***: high significance, ^ns^: not significant (*p* < 0.05).

**Table 6 ijerph-19-02029-t006:** Silica content (mg/kg tissue) in male rat liver, kidney, and spleen tissues 1, 3, and 6 days post-oral administration of 250 mg ≤ 50 nm TiSiO_4_ NPs/kg b.w.

Tissue	Si Concentration (mg/kg Tissue) at the Indicated Days after Exposure					
	1		3		6	
	Control	TiSiO_4_	Control	TiSiO_4_	Control	TiSiO_4_
Liver	8.01 ± 0.24	14.39 ± 2.21 **	9.00 ± 0.58	19.15 ± 1.00 ***	8.21 ± 1.66	8.61 ± 0.89 ^ns^
*p*		0.0076		0.0001		0.7307
Kidney	30.17 ± 1.89	44.74 ± 1.92 ***	30.19 ± 5.03	53.07 ± 1.36 **	22.46 ± 1.07	30.61 ± 2.11 **
*p*		0.0007		0.0016		0.004
Spleen	61.18 ± 4.45	67.62 ± 1.73 ^ns^	60.60 ± 4.82	87.14 ± 5.04 **	59.94 ± 3.17	65.40 ± 2.11 ^ns^
*p*		0.0798		0.0027		0.1663
Lung	31.62 ± 0.59	57.97 ± 2.48 ***	33.93 ± 2.52	65.07 ± 4.67 ***	32.45 ± 1.61	76.38 ± 3.10 ***
*p*		0.0001		0.0005		0.0002

Note: Values are the mean ± SD **: moderate significance, ***: high significance, ^ns^: not significant. (*p* < 0.05).

**Table 7 ijerph-19-02029-t007:** Titanium content (mg/kg tissue) in male rat liver, kidney, and spleen tissues after 1, 3, and 6 days post-oral administration of 250 mg ≤ 50 nm TiSiO_4_ NPs/kg b.w.

Tissue	Ti Concentration (mg/kg Tissue) at the Indicated Days after Exposure					
	1		3		6	
	Control	TiSiO_4_	Control	TiSiO_4_	Control	TiSiO_4_
Liver	<0.05	<0.05	<0.05	0.77 ± 0.02	<0.05	<0.05
Kidney	<0.05	<0.05	<0.05	2.42 ± 0.17	<0.05	<0.05
Spleen	<0.05	<0.05	<0.05	<0.05	<0.05	<0.05
Lung	<0.05	<0.05	<0.05	<0.05	<0.05	<0.05

Note: Values are the mean ± SD.

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
