# Peer review of "Potential Toxic Effects of Exposure to Titanium Silicon Oxide Nanoparticles in Male Rats"

_ijerph, 2022, doi:10.3390/ijerph19042029_

Round 1

Reviewer 1 Report

  1. What is the surface potential of TiSiO4 NPs? Please measure using the Zetasizer.
  2. There are various typos and unwanted usage of capital letters, and different font/styles were used. Please correct them to look even.
  3. The size of NPS should be carefully analyzed; mentioned ~ 50 nm is not correct. Authors need to calculate the average diameter of the NPs (minimum 100 individual particles) using ImageJ software and error calculation. As mentioned earlier, authors also need to analyze the size and charge (zeta potential) using Zetasizer. Please also assess the colloidal stability at least for 24 h.
  4. XRD analyses are not proper, and please assign all the peaks observed in the spectra and compare them with the literature. Please label the properly axes and use high-quality images.
  5. Please comment on the biodegradability of the TiSiO4 NPs in the rats after 28 days of exposure? Did authors carry any experiments to know their degradability in the different organs (liver, spleen, etc.)?

Author Response

Thank you very much for reviewing our manuscript . Your valuable comments and suggestions are highly appreciated .

Reviewer 2 Report

1) Did you test the toxicity of the particles in vitro?

2) How the 250 mg dose was selected? Why only one dose? Determine the limit dose where nanoparticles toxicity start could be contribution for this manuscript. 

3) Did you evaluate the serum biological markers at 24 hours? Why was only at long time points?

4) For the table 6, if a graphs can be done to compare better the biodistibution data. Why the Ti concentrations were low?

Author Response

Thank you for reviewing our manuscript . Your valuable comments and suggestions are highly appreciated .

Point 1: Did you test the toxicity of the particles in vitro?

Response 1: No we did not test the toxicity of the particles in vitro. However, , experiments are still required to determine the mechanism of harmful effects of TiSiO4 NPs in various cell lines.

Point 2: How the 250 mg dose was selected? Why only one dose? Determine the limit dose where nanoparticles toxicity start could be contribution for this manuscript. 

Response 2: Very few toxicological data exist for TiSiO4 NPs. Thus, there were no available data for LD50, NOAEL and LOAEL values, oral absorption, translocation, biodegradability and elimination of TiSiO4. Therefore, our study examined the response to an oral single dose of TiSiO4 NPs (250 mg /kg b.w.) in adult male rats at two-time post-administration (7 and 28 days after exposure. Along the experimental period in this study there were no recorded mortalities or observed toxicity symptoms among the treated rats. So, further low dose-response and long-term exposure studies are needed to evaluate the role of TiOSi4 NPs in potential toxicological effects and health hazards associated with oral exposure.

Point 3: Did you evaluate the serum biological markers at 24 hours? Why was only at long time points?

Response 3: No we did not evaluate the serum biological markers at 24 hours.The evaluation was only at long time points as there is noavailable data regarding absorption, distribution and translocation of TiOSi4 NPs after oral adminstration of a single dose.

 the initial absorption, to systemic sites is an important step in toxicokinetics.

Point 4: For the table 6, if a graphs can be done to compare better the biodistibution data. Why the Ti concentrations were low?

Response 4: The biodistibution data for Si and Ti are presented in tables because most Ti values are below the level of detection. Below detection limit of titanium in rats tissues may indicating breakdown of TiSiO4 NPs to Ti- products with low absorption, translocation and fast elimination. Besides, in this study accumulation of TiSiO4 NPs were not determined in the examined tissues.